# Effects of Two *Trichoderma* Strains on Apple Replant Disease Suppression and Plant Growth Stimulation

**DOI:** 10.3390/jof10110804

**Published:** 2024-11-20

**Authors:** Wen Du, Pengbo Dai, Mingyi Zhang, Guangzhu Yang, Wenjing Huang, Kuijing Liang, Bo Li, Keqiang Cao, Tongle Hu, Yanan Wang, Xianglong Meng, Shutong Wang

**Affiliations:** 1College of Plant Protection, Hebei Agricultural University, Baoding 071001, China; d_ven1@163.com (W.D.); daipengbo@hebau.edu.cn (P.D.); 18348934688@163.com (M.Z.); zhibaolibo@163.com (B.L.); cao_keqiang@163.com (K.C.); tonglemail@126.com (T.H.); wyn3215347@163.com (Y.W.); cugmxl@163.com (X.M.); 2State Key Laboratory of North China Crop Improvement and Regulation, Baoding 071001, China; 3Horticultural Research Institute Yunnan Academy of Agricultural Sciences, Kunming 650205, China; ygz@yaas.org.cn (G.Y.); angela_56@163.com (W.H.); 4College of Life Science, Hengshui University, Hengshui 053000, China; zwbh201011@163.com

**Keywords:** apple replant disease, *Fusarium oxysporum*, *Trichoderma* spp., biocontrol, colonization

## Abstract

*Fusarium oxysporum*, the pathogen responsible for apple replant disease (ARD), is seriously threatening the apple industry globally. We investigated the antagonistic properties of *Trichoderma* strains against *F. oxysporum* HS2, aiming to find a biological control solution to minimize the dependence on chemical pesticides. Two of the thirty-one *Trichoderma* strains assessed through plate confrontation assays, L7 (*Trichoderma atroviride*) and M19 (*T. longibrachiatum*), markedly inhibited = *F. oxysporum*, with inhibition rates of 86.02% and 86.72%, respectively. Applying 1 × 10^6^ spores/mL suspensions of these strains notably increased the disease resistance in embryonic mung bean roots. Strains L7 and M19 substantially protected *Malus robusta* Rehd apple rootstock from ARD; the plant height, stem diameter, leaf number, chlorophyll content, and defense enzyme activity were higher in the treated plants than in the controls in both greenhouse and field trials. The results of fluorescent labeling confirmed the effective colonization of these strains of the root soil, with the number of spores stabilizing over time. At 56 days after inoculation, the M19 and L7 spore counts in various soils confirmed their persistence. These results underscore the biocontrol potential of L7 and M19 against HS2, offering valuable insights into developing sustainable ARD management practices.

## 1. Introduction

Apple replant disease (ARD) impedes the growth and yield of newly planted apple trees, which is attributed to these soil microbes being introduced via cultivation practices. The ARD incidence has increased globally as a consequence of the rebuilding of orchards and the reuse of nursery stock [1]. Continuous cropping deteriorates the soil aggregate structure and alters the physical and chemical properties of the soil [2]. The prevalence of ARD is associated with the abundance and activity of soil microbial communities [3]. A substantial link exists between *Fusarium* species and ARD in China’s primary apple-producing regions, namely Bohai Bay and the Loess Plateau [4]. After infection and colonization with *F. oxysporum*, spores are produced in the roots and rhizomes of the plant, and the above-ground tissues are simultaneously damaged. *F. oxysporum* spores rapidly proliferate in the soil around the plant’s root system. As such, *F. oxysporum* is an important pathogenic fungus responsible for ARD [5].

Implementing correct fertilizer application, soil improvement methods, and grafting techniques facilitates the management of ARD. These methods increase the number of microorganisms in the soil and strengthen the microbial community structure and thus are important as strategies to control ARD. Simon et al. (2020) found that organic fertilizers possess the ability to combat apple root pathogens [6]. Biological control approaches, such as the use of *Lysobacillus capsici* AZ78, which releases volatile organic compounds (VOCs) that inhibit soilborne diseases, contribute to stabilizing and sustaining the soil system [7].

*Trichoderma* is a popular choice for plant disease management due to its adaptability and broad-spectrum disease resistance [8]. *Trichoderma* employs a range of biocontrol mechanisms against pathogens, including hyperparasitism, competition, and growth promotion [9,10,11]. Hyperparasitism is crucial in *Trichoderma*, involving the production of enzymes to combat other plant pathogens. This mechanism involves the use of chitinase, beta-glucanase, and protease to reduce the incidence of infections, particularly those occurring in the soil [12]. *Trichoderma* can penetrate or degrade the mycelia of pathogenic fungi, leading to cell breakdown and growth suppression [13]. *Trichoderma*’s competitive advantage lies in its rapid growth rate, which limits the nutrients and space available to pathogenic microorganisms. Its superior adaptability and growth enable *Trichoderma* to quickly outcompete pathogens [14], as demonstrated in tests against *F. oxysporum* [15]. *Trichoderma* competes for resources and promotes plant health by inhibiting diseases depending on the species and environmental conditions. The use of *Trichoderma* as a biofertilizer may result in increased crop yields and reduce the environmental impact of chemical fertilizer use in agriculture [16].

Commercial *Trichoderma* agents for soil improvement and disease management are becoming more popular because of their stability, high spore output, and quick development [17,18]. *Trichoderma asperellum* 6S-2 can control apple replanting disease. This strain increased the seedling growth and strengthened the soil microbial community structure, lowering the counts of *Fusarium* and other disease-causing fungi [19,20].

We identified *F. oxysporum* as the primary pathogen of ARD in our previous study. *F. oxysporum* strain HS2 was identified through phylogenetic analysis using the ribosomal DNA internal transcribed spacer (rDNA-ITS) and translation elongation factor 1α (EF-1α) sequences. Strain HS2 was most effective against ARD, producing the highest mortality rate, indicating its pathogenicity [21]. We explored the biocontrol potential of two *Trichoderma* strains, L7 and M19, both of which demonstrated strong inhibitory effects against *F. oxysporum*. Initial trials using mung beans helped with determining the optimal *Trichoderma* suspension concentrations, and subsequent greenhouse and field experiments with *M. robusta* Rehd apple rootstock validated *Trichoderma*’s biocontrol efficacy. The results of the field studies confirmed the colonization with *Trichoderma* isolates, establishing a foundation for using *Trichoderma* biocontrol to combat apple replant disease.

## 2. Materials and Methods

### 2.1. Fungal Strains and Plant Growth

The *Trichoderma* strains used in this study are detailed in Appendix A. After seven days of growth on potato dextrose agar (PDA) at 25 °C in the dark, the *Trichoderma* strains were washed with sterile 0.9% saline to collect spores. The spore suspension was prepared by rinsing the sample with sterilized 0.9% normal saline, which was followed by centrifugation at 6000 rpm at 25 °C for 5 min using an Eppendorf 5424R centrifuge. This process was repeated three times. The concentration of spores was adjusted to 1 × 10^7^ spores/mL using a hemocytometer (XB-K-25Plus, Shanghai Qiujing Biochemical Reagent & Instrument Co., Ltd., Shanghai, China) and prepared for immediate use. Initial pot experiments with the *Trichoderma* strains were conducted using the susceptible Ming mung bean variety. The seeds were soaked in sterile water and germinated for one day on moist sterile gauze until whitening. Subsequently, the seeds were sown, and, at the two-leaf stage (five days after germination), the mung beans were used for potted experiments. In the potted and field experiments, we used *M. robusta* Rehd as the apple rootstock. The robusta seeds were evaluated after reaching the four-leaf stage.

### 2.2. Determination of Plate Antagonistic Activity of Trichoderma Strains

Using HS2 as the target strain, plate confrontation assays were conducted with previously isolated *Trichoderma* strains. HS2 and *Trichoderma* strains were separately cultured on PDA medium for four days, and fungal discs (Φ5 mm) were obtained from each culture. Simultaneous inoculations were performed on both sides of a 90 mm diameter PDA Petri dish, with the fungal discs placed equidistant from the centerline. The plates were then incubated in the dark at 25 °C using an incubator (PRX-350C, Ningbo Haishu Saifu Laboratory Instrument Co., Ltd., Ningbo, China) for confrontation. The control involved inoculating only HS2, and each treatment was replicated three times. After five days, when the control reached three-quarters of the dish, the colony radius was measured, and the inhibition rate was calculated.
Inhibition Rate (%) = (Control Colony Radius (mm) − Treatment Colony Radius (mm)) × 100

### 2.3. Observation of Trichoderma Strains with a Scanning Electron Microscope

Glass slides were sterilized using an autoclave (SX-700, Tomy Digital Biology Co., Ltd., Tokyo Metropolis, Japan), dried, and placed in the center of 90 mm diameter Petri dishes. Five milliliters of PDA medium was poured onto each glass slide and allowed to solidify. Fungal discs (5 mm in diameter) of *Trichoderma* strains and HS2 were inoculated at opposite ends of the glass slide. A control dish was inoculated with HS2 only. The dishes were then incubated in the dark at 25 °C until the two fungi intersected.

Electron microscopy (SU8600 series, Servicebio, Wuhan, China) was employed to observe and photograph the interactions between *Trichoderma* and HS2. After five days of confrontation on PDA medium, the fungal blocks near the edge of the HS2 growth were selected for scanning electron microscopy observation and imaging.

### 2.4. Morphological Identification

*Trichoderma* strains were cultured on PDA medium and incubated in the dark at 25 °C using an incubator until the colonies covered the entire dish (approximately three days). Glass slides and cover slips were sterilized and handled in a laminar flow hood. PDA medium was applied to the glass slide, inoculated with *Trichoderma*, and covered with a sterilized cover slip. The dishes were then incubated for 24 h in darkness at 25 °C using an incubator. Optical microscopy (CX23, Sunny Optical Technology Co., Ltd., Ningbo, China) was used to observe and photograph the conidiophores and conidia morphology of *Trichoderma*.

### 2.5. Molecular Identification

The L7 and M19 strains were molecularly identified by amplifying the sequences for the internally transcribed spacer (ITS) region [22]. We utilized the CTAB method to extract the DNA from the L7 and M19 strains [23], which was followed by PCR amplification using a C1000 Touch PCR system (Bio-Rad, Shanghai, China). The L7 and M19 strains were molecularly identified using universal primers ITS1 (5′-TCCGTAGGTGAACCTGCGG-3′) and ITS4 (5′-TCCTCCGCTTATTGATATGC-3′). The PCR products were 598 and 626 bp for the L7 and M19 strains, respectively. The PCR reaction mixture (25 μL total volume) consisted of 12.5 μL of 2× Taq PCR Mix (product No. MF002-plus, Mei5Bio, Beijing, China), 1 μL each of the ITS1 and ITS4 primers, 9.5 μL of ddH2O, and 1 μL of DNA template. Amplification was performed under the following conditions: initial denaturation at 95 °C for 2 min, followed by 35 cycles of 94 °C for 40 s, 55 °C for 30 s, and 72 °C for 1 min, with a final extension at 72 °C for 7 min. After confirming the PCR product quality with electrophoresis, the amplified DNA was sequenced by Sangon Biotech, Shanghai, China The sequencing results underwent BLAST analysis for species determination. The ITS sequences of the same *Trichoderma* species were retrieved from the NCBI database. Phylogenetic analysis was performed using MEGA 6.0 software, employing the neighbor-joining method to construct a phylogenetic tree.

### 2.6. Radicle Inoculation Method Evaluating Biocontrol Against HS2 in Mung Bean Sprouts

The radicle inoculation method was used to assess the efficacy of *Trichoderma* spore suspensions at 1 × 10^5^, 1 × 10^6^, and 1 × 10^7^ spores/mL in preventing HS2 infection in mung bean sprouts. The controls were treated with sterile water only. Each treatment was replicated three times. After incubation in the dark at 25 °C for five days, mung bean embryo root lengths and disease incidence were investigated to determine the optimal concentration for the subsequent experiments.

### 2.7. Growth-Promoting Effect of Trichoderma Spore Suspension on Mung Beans

The mung bean seeds were soaked in a *Trichoderma* spore suspension (concentration: 1 × 10^6^ spores/mL) until germination. The plants were grown in a greenhouse at 25 °C, a relative humidity of 60%, and a 16 h light/8 h dark photoperiod. Using a syringe, 5 mL of the suspension was injected into the soil on both sides of the main root, approximately 1 cm away and 2–3 cm deep in the soil, resulting in a total root irrigation volume of 10 mL per 20 g of substrate. The treatments included seed soaking only and seed soaking combined with root irrigation. Watering was performed every two days. After ten days, the mung bean seedling height, fresh weight, and root length were measured; photographs were taken.

### 2.8. Evaluation of Trichoderma Spore Suspension Treatment on Octagonal Hawthorn Growth

*M. robusta* Rehd seeds were soaked in a *Trichoderma* spore suspension at the end of a sand storage period. Sterile water treatment served as the control. After germination, *M. robusta* Rehd seedling roots were irrigated with the *Trichoderma* spore suspension (30 mL per plant). Watering occurred every two days, and root irrigation with the biocontrol agent was performed every ten days. After thirty days, *M. robusta* Rehd seedling height, fresh weight, chlorophyll content (measured using a SPAD-502Plus, Top Yun Agricultural Instrument Co., Ltd., Hangzhou, China), root weight, root length, and enzyme activity were investigated.

The defense enzyme activities were measured following the manufacturer’s instructions (Solarbio, Beijing, China) for CAT (Cat. No. BC0205), SOD (Cat. No. BC0170), and PAL (Cat. No. BC0210) assay kit, all of which were determined via visible spectrophotometry.

### 2.9. Protective Effect of Trichoderma on Apple Replant Disease Caused by F. oxysporum

The *M. robusta* Rehd seeds underwent germination and seeding as described above. The controls included separate inoculation with pathogenic soil and water. Each treatment consisted of four *M. robusta* seedlings, with three repetitions per treatment. After thirty days, plant height, fresh weight, chlorophyll content, root weight, root length, and enzyme activity were measured. The disease index was measured using modified disease grading standards [24]; the criteria were as follows: 0 = asymptomatic; 1 = yellowing leaves with less than 1/8 of the leaves affected; 2 = yellowing leaves, stunted plants, with 1/8 to 1/4 of the leaves affected; 3 = yellowing leaves, stunted plants, with brown roots and 1/4 to 1/2 of the leaves affected; 4 = wilting, drying, and root necrosis, with more than 1/2 of the leaves affected.

### 2.10. Field Experiments

Field experiments were conducted at the National Apple Industry Technology System Disease and Pest Control Experimental Orchard of Hebei Agricultural University. *M. robusta* Rehd seedling plots were categorized into two groups: normal and continuous cropping soil. Normal cropping soil referred to soil where apple rootstock was planted for the first time, and continuous cropping soil referred to the continuous cropping of apple rootstock in the orchard. The weeds were removed from the plots, which were plowed in preparation for the experiments. Healthy and uniformly growing *M. robusta* Rehd seedlings at the four-leaf stage, previously cultivated in the greenhouse, were transplanted into the field plots. Seedling roots were irrigated within 24 h of transplantation. The roots of the *M. robusta* Rehd seedlings were irrigated with one of the two strains of *Trichoderma* spore in the experimental orchard. Each treatment included nine *M. robusta* Rehd seedlings, with negative controls receiving water-only treatments. Roots were irrigated every ten days, and measurements of plant height, stem diameter, and chlorophyll content were recorded at the 60th day in both the continuous and rotation cropping plots.

### 2.11. Fluorescent-Labeled Trichoderma Strains

We followed a previously reported method for the fluorescence labeling test of the L7 and M19 strains [25]. First, the L7 and M19 protoplasts were prepared with 0.6 mol/L MgSO_4_·7H_2_O as the mother solution, a lysozyme concentration of 5 mg/mL, a mycelium culture time of 16 h, an enzyme digestion time of 3.5 h, and a protoplast concentration of 5 × 10^6^/mL. Second, PEG-mediated genetic transformation was performed, 200 μL of protoplast was mixed with PCT74 plasmids, and the plasmids were left on ice for 20 min. Third, we added PTC solution drop by drop, which was left on ice for 20 min. Finally, TB3 medium was added and oscillated at 25 °C and 150 r/min for 24 h. After vortex mixing the mycelium, 50 mL of bottom medium was added. After solidification, the mycelium was placed in a constant-temperature incubator at 25 °C for 10 h, and then top medium containing hygromycin was added. The transformants were selected and cultured in PDA medium with or without hygromycin resistance for 5 generations to ensure the genetic stability of the transformants. Fluorescence was observed under a fluorescence microscope (ECLIPSE Ti2-U, Nikon, Shanghai, China) to select the transformants with strong fluorescence. The DNA of the strain was extracted for PCR verification.

### 2.12. Field Colonization Capacity Test

The spore suspensions of the two fluorescently labeled *Trichoderma* strains were adjusted to 1 × 10^6^ spores/mL. Each *M. robusta* Rehd seedling’s roots were irrigated with 300 mL of the spore suspension. Treatments included three *M. robusta* Rehd seedlings per group, with the control group receiving 300 mL of water as a treatment. Immediately after treatment, soil samples were collected around the root irrigation site under clear-weather conditions. Regular watering was conducted, and soil samples were obtained every seven days after root irrigation. Soil samples were processed: one gram of air-dried soil was added to a flask with sterile water, allowed to settle, and then aspirated onto a glass slide for microscopic observation of the labeled *Trichoderma* presence. Additionally, 100 μL of the suspension was plated on PDA agar plates containing gentamicin and ampicillin, spread evenly, and incubated at 25 °C in the dark for 36 h in an incubator. Colonies were counted, and the number of colonies per gram of soil sample was calculated using the following formula:Colonies per gram of soil sample = (C ÷ V) × M
where (C) is the average number of colonies that grew on the plate at a certain dilution, (V) is the volume of the dilution solution used for spreading (mL), and M is the dilution factor.

### 2.13. Data Analysis

After organizing the experimental data from the radicle inoculation test, the growth indicators of *M. robusta* Rehd seedlings, and the antifungal activity of *Trichoderma* spp. using Microsoft Excel 2021, statistical analysis was conducted using SPSS 22.0. Single-factor analysis of variance was performed, and significant differences were determined using the LSD and Duncan’s new complex range tests at confidence levels of *p* < 0.05 and *p* < 0.1. Graphs were created using GraphPad Prism 9.0, and images were edited using Adobe Photoshop 2021.

## 3. Results

### 3.1. Screening and Characterization of Biocontrol Trichoderma Strains Exhibiting High Inhibitory Effects Against F. oxysporum

The inhibitory effect of *Trichoderma* on HS2 was evaluated using the plate confrontation method. Strains M19 and L7 achieved inhibition rates of over 86% against HS2 and produced a large number of spores (Table 1).

The observations from the Petri dish assays revealed a distinct light yellow antagonistic zone at the mycelial intersection, where *Trichoderma* gradually covered HS2 (Figure 1). Microscopic examination showed the coiling of *Trichoderma* strains around the mycelia of HS2 (Figure 1). Scanning electron microscopy further demonstrated that the two strains caused the twisting, collapsing, and rupturing of the HS2 mycelia during the parasitism process (Figure 1).

These results indicate that strains M19 and L7 are highly effective in inhibiting the growth of HS2, suggesting their potential as biocontrol agents against *F. oxysporum*.

### 3.2. Trichoderma Strains Were Identified Based on Morphological and Molecular Characteristics

Strain L7 formed circular and velvety colonies, with light green conidia and slender mycelia. The phialides were slender, and the conidia were nearly spherical or ovoid, with lengths of 3.0–4.5 μm and widths of 2.5–4.0 μm. Strain M19 had light green conidia, with colonies radiating outward from the center, showing centrally high sporulation rates. The mycelia exhibited arborescent branching and were dendritic, with ovate conidia measuring 2.0–6.0 μm in length and 2.0–3.0 μm in width (Figure 2A).

Based on the phylogenetic analysis, strain M19 clustered within the *T. longibrachiatum* branch, and strain L7 clustered within the *T. atroviride* branch. Consequently, strain M19 was identified as *T. longibrachiatum* and strain L7 as *T. atroviride* (Figure 2B).

### 3.3. Screening of Spore Suspensions of Trichoderma Strains and Their Growth-Promoting Effects on Mung Beans

In the mung bean root inoculation experiment, the optimal growth promotion effect was observed when the spore suspension concentration of the two *Trichoderma* strains was 1 × 10^6^ spores/mL. After seeding, compared with the control (CK), the roots were over 50% longer, the number of fibrous roots was over 20% larger, and the disease incidence significantly decreased (Appendix A). As such, a spore suspension concentration of 1 × 10^6^ spores/mL was used in the subsequent experiments.

The mung bean seedlings were treated with the spore suspensions for 10 days to evaluate the growth-promotion effects of strains M19 and L7. *Trichoderma* treatment increased plant height, fresh weight, dry weight, and root length. Notably, root irrigation following seed soaking promoted growth the most. The M19 treatment resulted in increases of 160.38% in plant height, 81.42% in fresh weight, 329.43% in dry weight, and 33.73% in root length. Similarly, the L7 treatment led to increases of 161.41% in plant height, 78.66% in fresh weight, 345.33% in dry weight, and 32.66% in root length (Appendix A). Consequently, in the subsequent experiments, we employed seed soaking combined with root irrigation.

### 3.4. Growth-Promotion Effect of Trichoderma Spore Suspension on M. robusta Rehd Plants in Pot Experiments

The potted experiments were conducted using the previously determined concentration and processing method to study the growth of *M. robusta* Rehd seedlings. The results showed that the method strongly promoted growth. The plant height increased by 69.53% with strain M19 and 48.22% with strain L7. The roots were longer by over 21% with strains M19 and L7 compared with the control (CK). The fresh plant and root weight increased by 77.92% and 99.58%, respectively, when treated with strain M19, and by 104.50% and 105.37% when treated with strain L7, respectively. The leaf number increased by 24.10% and 8.43% when treated with strains M19 and L7, respectively, compared with the control. The chlorophyll content increased by more than 155% compared with that in the control (Figure 3).

Treatment with *Trichoderma* strains L7 and M19 substantially increased the activities of various defense enzymes and the root vitality in *M. robusta* Rehd seedlings. Specifically, the activity of phenylalanine ammonia-lyase (PAL) increased by 123.16% with L7 and by 118.95% with M19 compared with the control. The catalase (CAT) activity increased 134.15% with L7 and 139.02% with M19. The superoxide dismutase (SOD) activity rose by 299.85% with L7 and 268.33% with M19 compared with the control. Additionally, the root vitality increased by 161.70% with L7 and by 153.98% with M19 (Figure 4). These findings indicate that *Trichoderma* treatments significantly increased the activity of defense enzymes (PAL, CAT, and SOD) and the root vitality of *M. robusta* Rehd seedlings, leading to improved plant health and stress resistance.

### 3.5. Control Effect of Trichoderma on M. robusta Rehd Plants in Greenhouse Experiment

Greenhouse experiments were conducted to investigate the effects of *Trichoderma* strains M19 and L7 on controlling replant disease in *M. robusta* Rehd seedlings caused by *F. oxysporum*. The seedlings treated with HS2 exhibited wilting and death, whereas more than 87% of those treated with spore suspensions of strains M19 and L7 were protected more than 87% against the disease (Table 2). The seedlings treated with strains L7 and M19 grew significantly more than those treated with *F. oxysporum* alone. Specifically, the plant height increased by 79.88% with strain L7 and by 94.84% with strain M19. The root length was increased 30.98% and 31.95% with strains L7 and M19, respectively. The fresh root weight increased by 185.12% with strain L7 and by 182.74% with strain M19. The number of leaves increased by 62.99% and 61.69% for strains L7 and M19, respectively, whereas the chlorophyll content rose by 19.05% with L7 and by 15.56% with M19 compared with the control. These *Trichoderma* spore suspension treatments of diseased soil markedly increased the plant height, fresh weight, root length, fresh root weight, leaf number, and chlorophyll content of *M. robusta* Rehd seedlings (Appendix A).

The activities of the defense enzymes (SOD and CAT) and root vitality were stronger in the seedlings treated with *Trichoderma* compared with those treated only with *F. oxysporum* (Appendix A). These findings demonstrated that strains M19 and L7 effectively protected *M. robusta* Rehd seedlings from pathogen invasion, ensuring favorable growth conditions and serving as potential antagonists for controlling ARD.

### 3.6. Biocontrol Efficacy of Trichoderma on M. robusta Rehd Plants in Field Experiment

The results of the field experiments demonstrated that treatments with *Trichoderma* strains M19 and L7 considerably increased the growth of *M. robusta* Rehd planted in both normal and continuous cropping soils.

In the normal cropping soil, the plant height, stem diameter, leaf number, and chlorophyll content of *M. robusta* Rehd seedlings with the L7 treatment were increased by 95.37%, 31%, 36.90%, and 6.02%, respectively, compared with the control treatment. After the M19 treatment, these values increased by 165.82%, 36.01%, 61.90%, and 23.44%, respectively (Figure 5).

In continuous cropping soil, after the L7 treatment, the plant height, stem diameter, leaf number, and chlorophyll content of the *M. robusta* Rehd seedlings increased by 75.17%, 62.72%, 29.82%, and 16.80%, respectively, compared with the control. After the M19 treatment, these values increased by 62.24%, 68.70%, 48.25%, and 11.71%, respectively (Figure 6).

### 3.7. Acquisition of Trichoderma GFP-Labeled Strains and Determination of Field Colonization Ability with M. robusta Rehd Plants

#### 3.7.1. Construction of Fluorescent-Labeled Strains

The transformants exhibited no morphological differences from the wild type. PCR amplification confirmed the successful introduction of the GFP gene into the M19 and L7 strains (Appendix A).

#### 3.7.2. Field Colonization Capability of *Trichoderma* Strains

After root irrigation, the soil samples were obtained, diluted 100 times, and observed under a fluorescence microscope. The fluorescence signals were detected, confirming the presence of the transformed offspring carrying the GFP gene in the soil (Figure 7A). The initial spores were counted immediately after irrigation, demonstrating that the labeled strains had successfully colonized the *M. robusta* Rehd plants in both normal and continuous cropping soils (Figure 7B). Over time, the spore numbers of the labeled strains in the soil fluctuated before stabilizing, with the spore counts of strain M19 being consistently higher than those of strain L7 under both planting conditions. The results of fluorescence microscopy further confirmed the presence of the GFP-labeled transformants in the soil.

These results indicated that the GFP-labeled *Trichoderma* strains M19 and L7 successfully colonized the rhizosphere of *M. robusta* Rehd plants under field conditions, demonstrating their potential for application in biological control and plant growth promotion.

## 4. Discussion

Screening methods for biocontrol agents, such as plate confrontation, spore germination, and dual-culture techniques, are used to assess the inhibitory effects of these agents on pathogens by studying mutual competition suppression, volatile metabolite inhibition, and spore germination inhibition. The antagonistic effects of *Trichoderma* in various crop tests have been well documented, particularly against pathogens such as *Phytophthora infestans* and *F. oxysporum*, which share a hemibiotrophic lifestyle. Certain strains of *Trichoderma*, such as TL1-2A, have achieved substantial disease reduction and plant growth promotion, achieving a 40.2% disease mitigation effect in 2022–2023, highlighting their potential in sustainable plant disease management [26]. The research findings have emphasized the ecological influence of foliar treatments on maize fungal communities such as *F. verticillioides*. We identified *T. atroviride* and *T. longibrachiatum* as promising antagonistic strains using the plate confrontation method, which shows the potential for improving the management of apple replant disease.

The current studies suggest that *Trichoderma* fungi can boost plant growth and disease resistance, increasing output. Al-Naemi et al. (2016) grew and identified *T. harzianum* strains and investigated their pathogen suppression abilities. As a biocontrol agent, *T. harzianum* can diminish *Ceratocystis radicicola* necrosis by creating volatile metabolic products and producing cell-wall-degrading enzymes [27]. Hewedy et al. (2020) examined the genetic diversity and biocontrol efficiency of 15 indigenous *Trichoderma* strains from Egypt against the *Fusarium* wilt of pepper, which is caused by *F. oxysporum* f. sp. *capsici*. The results of the ITS sequencing analysis identified four *Trichoderma* species: *T. harzianum*, *T. asperellum*, *T. longibrachiatum*, and *T. viride*. In vitro and in planta, these strains strongly inhibited pathogen growth and promoted plant growth. Yu et al. (2021) highlighted the value of *T. atroviride* in enhancing the plant defense mechanisms and increasing the absorption of the available nitrogen in tomato seedlings, indicating its potential as a biocontrol agent for agricultural and forestry applications [28]. Degani et al. (2021) found that *T. longibrachiatum* can effectively reduce the late wilt disease symptoms in maize, significantly decrease pathogen levels, and increase plant growth [29]. Most *Trichoderma* fungal formulation research has been conducted in greenhouses or laboratories, requiring field trials to confirm the results [30]. We examined *M. robusta* Rehd, a common Northern Chinese apple rootstock used for apple seedling production. Two *Trichoderma* fungi were tested for disease resistance and growth promotion using in vivo embryo root inoculation, pot tests, and field trials. The results of the pot studies showed that the *Trichoderma* treatments considerably increased the *M. robusta* Rehd seedling growth. Most of the seedlings treated with *F. oxysporum* wilted in our disease prevention studies, whereas the M19 and L7 spore suspensions protected over 87% of the seedlings. In regular and continuous cropping soils, the *M. robusta* Rehd seedlings grew well with the *Trichoderma* spore suspension treatments. We hypothesized that strains M19 and L7 can serve as biocontrol agents to improve the *M. robusta* Rehd plant development and health in diverse soil conditions. This is based on our observation of the notable increases in plant height, stem diameter, leaf number, and chlorophyll content compared with the control.

Recent studies have demonstrated the effectiveness of *Trichoderma* strains in enhancing plant defense enzyme activities, which contribute to improved plant health and resistance against various pathogens. For example, treatment with *Trichoderma* strains can considerably enhance the activities of key defense-related enzymes, including phenylalanine ammonia-lyase (PAL), catalase (CAT), and peroxidase (POD). These enzymes play a pivotal role in plant defense mechanisms, assisting in the mitigation of stress and pathogen attacks. Diverse *Trichoderma* strains can effectively mitigate the effects of bacterial blight in rice by augmenting the activities of pivotal defense-related enzymes, including PAL, PPO, CAT, and POD. Furthermore, these strains can upregulate a number of defense-related genes, which supports their use as biocontrol agents in agricultural practices [31,32]. The two *Trichoderma* strains identified in our study significantly enhanced plant defense enzyme activity, thereby supporting their use as biological control agents. These findings contribute to the development of sustainable agricultural practices by enhancing plant defense mechanisms and improving plant health.

A comprehensive understanding of the colonization patterns of *Trichoderma* is essential for the successful commercial production of this organism. Stummer et al. (2020) investigated the colonization and persistence of three *Trichoderma* strains (*T. gamsii*, *T. afroharzianum*, and *T. harzianum*) in agricultural and wheat rhizosphere soils. Additionally, the efficacy of these strains against *F. pseudograminearum* was assessed. The results of quantitative PCR (qPCR) for the detection and monitoring of these strains indicated increased wheat biomass, colonization, and pathogen suppression [33]. By employing molecular tagging, Xian et al. (2020) traced the dispersion and survival of *T. asperellum* T4 within the soil, which demonstrated its capacity for long-term colonization and survival [34]. The results of our study confirm the efficacy of *T. asperellum* as a biocontrol agent and demonstrate the value of molecular tagging for evaluating the soil colonization of biocontrol organisms. In our study, fluorescence labeling and resistance marking were employed to monitor the colonization of *Trichoderma* strains in the field for a period of 56 days. Resistance marking was employed to count the *Trichoderma* spores, and fluorescence was used to test the colonization potential. The spore counts of the marked strains fluctuated before stabilizing over time. Both *Trichoderma* strains demonstrated robust environmental adaptation and stability in the field. Strain M19 consistently exhibited higher colonization spore counts than strain L7 in both normally cropped and continuously cropped soil, indicating that the soil conditions and strain type strongly influence *Trichoderma* colonization. These findings indicate that *Trichoderma* colonization is influenced by the soil environment and varies according to strain type.

The promotion of *Trichoderma* colonization through the adjustment of soil conditions could enhance agricultural production, supporting future industrialization and commercialization efforts. Effective colonization is a prerequisite for *Trichoderma* to exert its biocontrol effects. This study was limited in scope, focusing exclusively on the soil colonization of two *Trichoderma* strains without examining their interactions with pathogens, plants, or the soil environment. Furthermore, the 56-day monitoring period was limited; longer-term colonization requires further investigation.

## 5. Conclusions

Of the thirty-one *Trichoderma* strains screened, two were selected for their strong inhibitory effects against *F. oxysporum*, the pathogen responsible for apple replant disease. Our morphological and molecular characterization identified strain M19 as *T. longibrachiatum* and L7 as *T. atroviride*. These biocontrol strains promoted mung bean and *M. robusta* Rehd seedling growth. They also increased the *M. robusta* Rehd seedling root viability, as well as SOD, CAT, and PAL activities. These effects prevented *F. oxysporum* invasion and promoted seedling development. Fluorescent strains M19 and L7 colonized the root-zone soil of *M. robusta* Rehd seedlings for 56 days, with the colonization sustained at more than 37% of the original spore count. These microorganisms’ persistent colonization indicates their long-term biocontrol potential.

In the context of biocontrol applications, we must investigate the optimal application methods and environments for *Trichoderma* formulations, including their potential synergistic effects with other beneficial microorganisms. Maximizing the efficiency of biological disease control is crucial not only for ensuring the safety of agricultural products and advancing sustainable practices in China but also for contributing to the global efforts to reduce the reliance on chemical pesticides. The findings from our study can serve as a foundation for similar biocontrol strategies in other regions, especially those facing challenges with soilborne pathogens.

## Figures and Tables

**Figure 1 jof-10-00804-f001:**
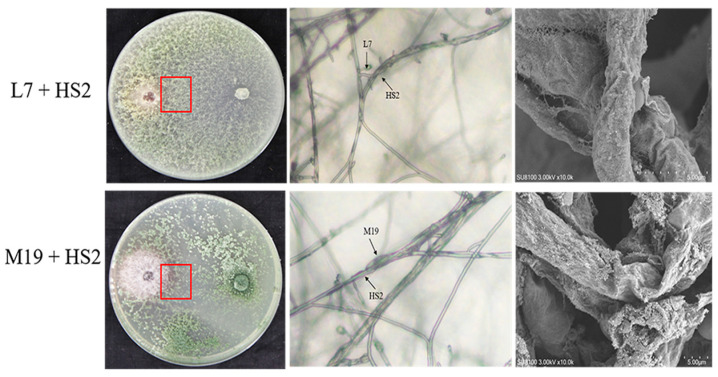
Antagonistic observation of biocontrol *Trichoderma* strains L7 and M19 against *F. oxysporum* HS2. The morphology of L7 and M19 Petri dishes is observed on the (**left**), showing a distinct light yellow antagonistic zone forming at the mycelial intersection, with *Trichoderma* gradually covering *F. oxysporum* HS2. The red boxes indicate the confrontation observation zones. In the (**middle**), microscopic observation reveals that the test strains cause twisting, collapsing, and rupturing of HS2 mycelia during the parasitism process. On the (**right**), scanning electron microscope images show *Trichoderma* strains coiling around the mycelia of HS2.

**Figure 2 jof-10-00804-f002:**
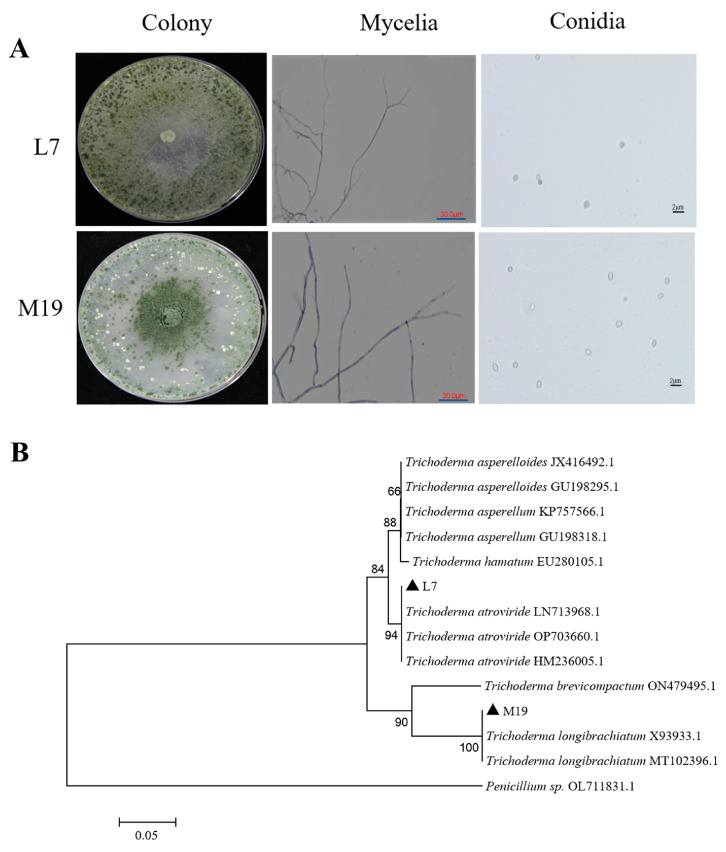
Identification of the tested *Trichoderma* isolates. (**A**) Colony morphology and microscopic observation of the tested *Trichoderma* isolates. L7 has circular and velvety colonies with light green conidia and slender mycelia. The phialides are slender, and the conidia are nearly spherical or ovoid, measuring 3.0–4.5 μm and 2.5–4.0 μm. M19 exhibits light green conidia with colonies radiating outward from the center, showing high sporulation rates centrally. The mycelia are tree-like, and the oval-shaped conidia measure 2.0–3.0 μm and 2.0–6.0 μm. (**B**) Phylogenetic trees of two *Trichoderma* strains constructed based on ITS sequences. Phylogenetic tree constructed by the neighbor-joining method based on ITS sequences. The percentage of replicate trees in which the associated taxa clustered together in the bootstrap test (1000 replicates) is shown next to the branches. The tree is drawn to scale, with branch lengths in the same units as those of the evolutionary distances used to infer the phylogenetic tree. The evolutionary distances were computed using the Poisson correction method and are in the units of the number of amino acid substitutions per site. Based on the tree, strain M19 clustered within the *T. longibrachiatum* branch, while L7 clustered within the *T. atroviride* branch. All positions containing gaps and missing data were eliminated. Evolutionary analyses were conducted in MEGA6.

**Figure 3 jof-10-00804-f003:**
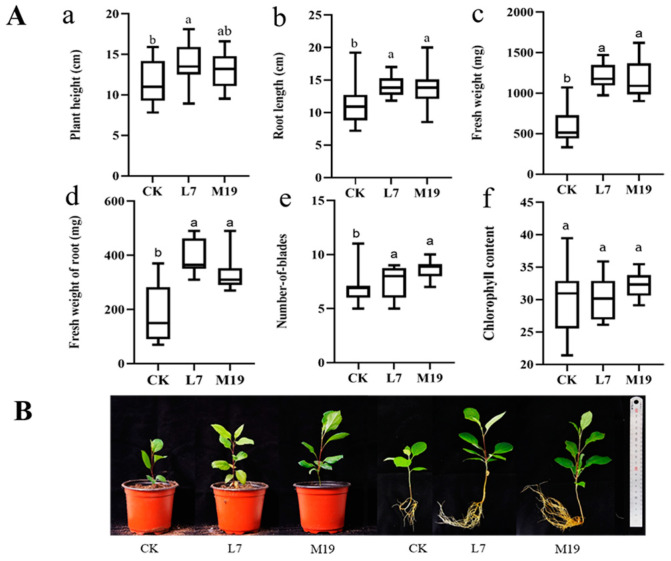
Effect of *Trichoderma* on the growth of *M. robusta* Rehd seedlings. (**A**) Determination of growth indexes of *Trichoderma* on *M. robusta* Rehd. Labels (**a**–**f**) represent seedling height, root length, fresh weight, root fresh weight, leaf number, and chlorophyll content, respectively. Values with superscript letters a and b are significanty diferent across columns (*p* < 0.05). Results showed significant improvements in *M. robusta* Rehd seedling parameters after treatment with strains M19 and L7 compared to the control (CK). (**B**) The effect of *Trichoderma* on the growth of *M. robusta* Rehd. CK represents *M. robusta* Rehd seedlings treated with only water, L7 represents seedlings treated with L7 spore suspension, and M19 represents seedlings treated with M19 spore suspension.

**Figure 4 jof-10-00804-f004:**
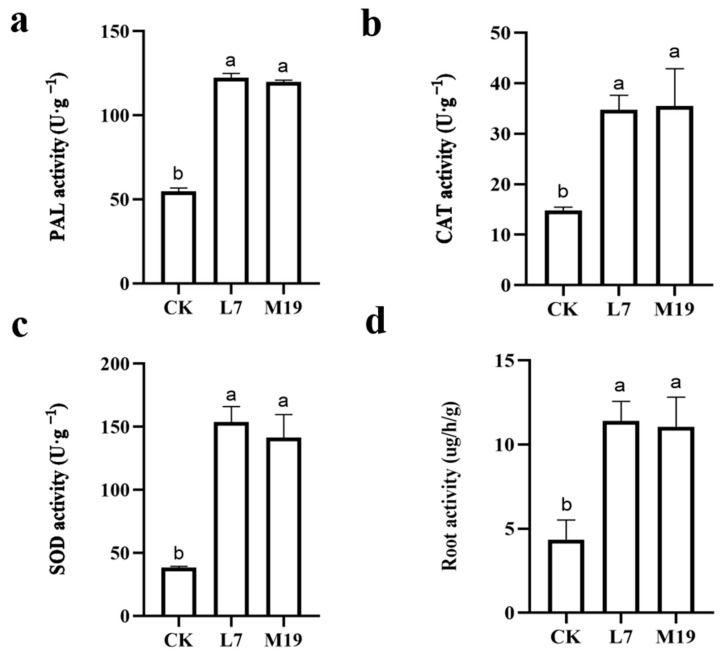
The effect of *Trichoderma* on the activity of defense enzymes in the roots of *M. robusta* seedlings. (**a**) SOD activity, (**b**) CAT activity, (**c**) PAL activity, and (**d**) root activity. CAT activity, SOD activity, PAL activity, and root vitality were all higher in *M. robusta* Rehd seedlings treated with the two *Trichoderma* strains compared to CK. Values with superscript letters a and b are significanty diferent across columns (*p* < 0.05).

**Figure 5 jof-10-00804-f005:**
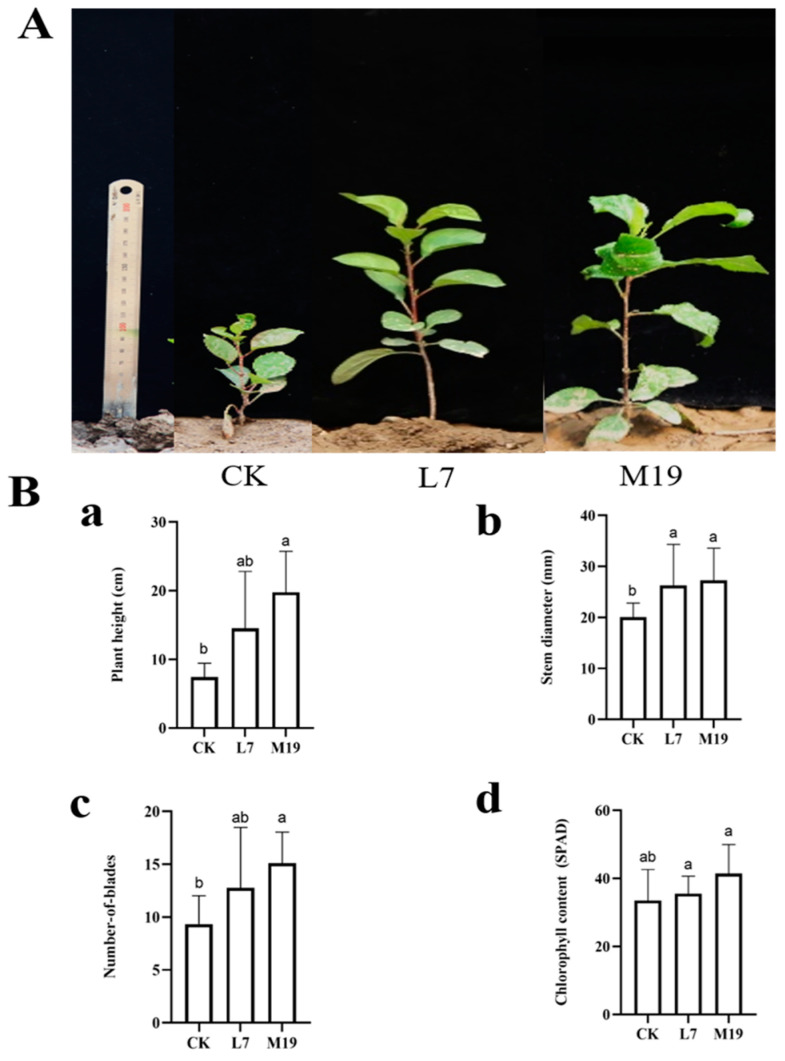
Effect of *Trichoderma* on *M. robusta* Rehd seedlings in normal cropping soil (60 days). (**A**) Growth of *M. robusta* Rehd seedlings in normal cropping soil for 60 days. CK represents *M. robusta* Rehd seedlings treated with only water, L7 represents seedlings treated with L7 spore suspension, and M19 represents seedlings treated with M19 spore suspension. The treatment of *Trichoderma* spore suspension in normal cropping soil significantly increased seedling height and demonstrated a strong growth-promoting effect. (**B**) Determination of physiological indexes of *M. robusta* Rehd seedlings growing in normal cropping soil for 60 days. Labels (**a**–**d**) represent seedling height, stem diameter, chlorophyll content, and leaf number, respectively. Values with superscript letters a and b are significanty diferent across columns (*p* < 0.05). Significant enhancements in seedling height, leaf number, chlorophyll content, and root health were noted, indicating a strong growth-promoting effect.

**Figure 6 jof-10-00804-f006:**
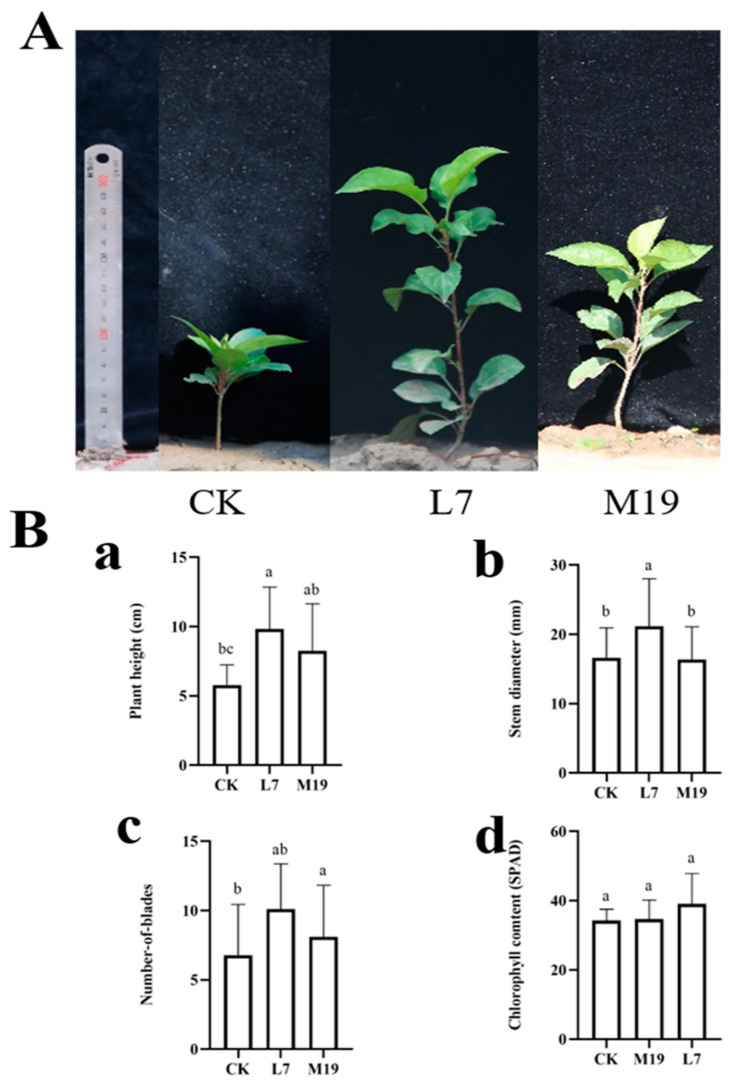
Effect of *Trichoderma* on *M. robusta* Rehd seedlings in continuous cropping soil (60 days). (**A**) Growth of *M. robusta* Rehd plants in continuous cropping soil for 60 days. CK represents *M. robusta* Rehd seedlings treated with only water, L7 represents seedlings treated with L7 spore suspension, and M19 represents seedlings treated with M19 spore suspension. The treatment of *Trichoderma* spore suspension in continuous cropping soil significantly increased seedling height and demonstrated a strong growth-promoting effect. (**B**) Determination of physiological indexes of *M. robusta* Rehd seedlings growing in continuous cropping soil for 60 days. Labels (**a**–**d**) represent seedling height, stem diameter, chlorophyll content, and leaf number, respectively. Values with superscript letters a, b and c are significanty diferent across columns (*p* < 0.05).

**Figure 7 jof-10-00804-f007:**
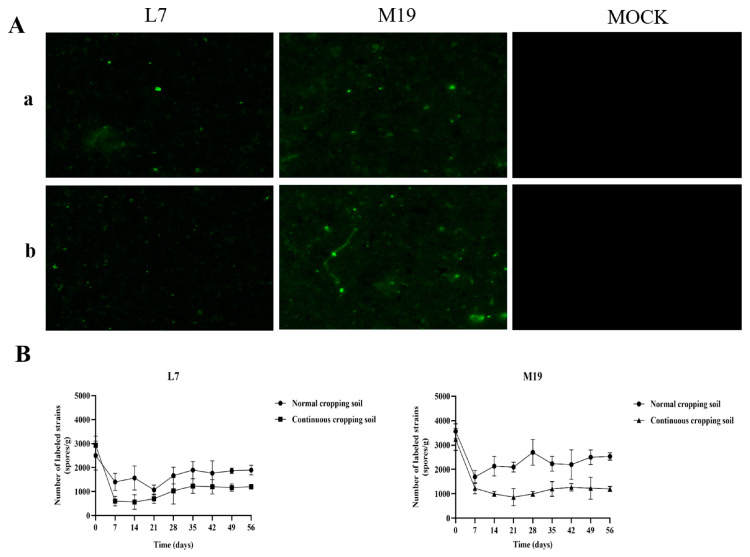
Fluorescence observation and colonization status of two transformants in soil suspension and *M. robusta* Rehd root soil. (**A**) Fluorescence observed by two transformants in soil suspension. (**a**) Represents normal cropping soil; (**b**) represents continuous cropping soil; L7, M19, and MOCK are fluorescence of L7 transformant in soil, fluorescence of M19 transformant in soil, and CK of soil. Samples were taken after root drenching treatment, diluted 100 times, and fluorescence was observed under a fluorescence microscope. (**B**) Colonization status of two transformants in the soil of *M. robusta* Rehd root. Over time, the spore counts of the marked strains fluctuated before stabilizing. Notably, the colonization spore count of strain M19 was higher than that of strain L7 in both soil types.

**Table 1 jof-10-00804-t001:** Inhibition effect of some tested *Trichoderma* strains on *F. oxysporum*.

Number	*Trichoderma* Strain	Colony Radius (cm)	Inhibition Rate (%)
1	L7	0.63 ± 0.03 **	86.72
2	M19	0.67 ± 0.03 **	86.02
3	L12	0.93 ± 0.03 *	80.43
4	T3-1	0.93 ± 0.03 *	80.43
5	T1-17	0.97 ± 0.07 *	79.73
6	T3-19	1.03 ± 0.09 *	78.34
7	M13	1.27 ± 0.15 *	73.45
8	M23	1.27 ± 0.09 *	73.45
9	T1-20	1.27 ± 0.09 *	73.45
10	T2-7	1.30 ± 0.10 *	72.75
11	T1-24	1.50 ± 0.10 *	68.55
12	T1-8	1.67 ± 0.03 *	65.06
13	T3-11	1.67 ± 0.18 *	65.06
14	T3-23	1.83 ± 0.09 *	61.57
15	T1-11	1.87 ± 0.09 *	60.87
16	T1-10	1.90 ± 0.06 *	60.17
17	L10	1.90 ± 0.26 *	60.17
18	M7	1.90 ± 0.10 *	60.17
19	M18-1	1.93 ± 0.12 *	59.47
20	T2-1	2.03 ± 0.03 *	57.37
21	T3-38	2.03 ± 0.09 *	57.37
22	T3-20	2.07 ± 0.07 *	56.67
23	T4-5	2.10 ± 0.06 *	55.97
24	T4-3	2.13 ± 0.09 *	55.28
25	L2	2.17 ± 0.09 *	54.58
26	T2-13	2.17 ± 0.09 *	54.58
27	M1	2.20 ± 0.10 *	53.88
28	T3-4	2.27 ± 0.15 *	52.48
29	T3-6	2.27 ± 0.15 *	52.48
30	L15	2.30 ± 0.06 *	51.78
31	T4-9	2.57 ± 0.15 *	46.19
32	CK	4.77 ± 0.03	0.00

Data as mean ± standard deviation. * represents significant difference from CK, ** represents significant difference from CK and other strains at *p* < 0.05 based on the Duncan range test.

**Table 2 jof-10-00804-t002:** Protective effect of *Trichoderma* on replant diseases caused by *F. oxysporum* HS2.

Treatment	Disease Index	Protection Effect
HS2	84.33 ± 4.91 ^a^	-
M19 + HS2	10.33 ± 2.96 ^b^	87.78
L7 + HS2	10.33 ± 2.96 ^b^	87.78

Note: Values with superscript letters a and b are significanty diferent across columns (*p* < 0.05).

## Data Availability

The study did not report any data.

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
