# Peer review of "Effects of Two Trichoderma Strains on Apple Replant Disease Suppression and Plant Growth Stimulation"

_jof, 2024, doi:10.3390/jof10110804_

Round 1
Reviewer 1 Report
Biocontrol agents have been considered as an efficeint and eco-friendly alternative to the use of fungicides to control plant-pathogenic fungi. Trichoderma sp. are among the most intensively used biocontrol tools for plant disease management. Therefore, the paper looks timely and scientifically significant. However, before being accdepted, it should be seriously improved according to the following comments:
1. The authors should clarify how many strains were actually analyzed in the study. In Abstract they declare that two most efficient strain were chosen from 62 initially tested ones. In Table 1, only 32 strains are mentioned. I believe that all information about 62 strains can be reported in a separate table, that would include they taxonomic identification, origin/collection, species identification and biocontrol effect. Also it would be interesting to know if these 62 strains were identified by morphological/molecular means. Moreover, the authors can add some information regarding T. atroviride and T. longibrachiatum species themselves. Have these species been used as biocontrol agents before?
2. More information about F. oxysporum HS2 strain is needed. What is its origin? How its species identification was confirmed? How its pathogenicity was determined? Also, the Introduction should contain more information about F. oxysporum as the main causal agent of ARD.
3. More information about molecular identification procedures is needed. What primers were used for identification? What were amplification conditions? PCR products size? Also, the authors could use more than a single DNA marker for identification (for instance, TEF1, beta-tubulin or calmodulin - this proposal is optional).
4. How were enzyme activities estimated?
Line 66: 'Previous results' instead of 'Researchs'
Line 71: Reference for a 'previous study'?
Line 72: 'F. oxysporum' in Italics
Line 84-85: How the spore were preserved?
Line 118: What electron microscope was used?
Line 127: What optical microscope was used?
Line 130: 'Molecular identification' instead of 'Molecular biology identification'
Line 132: Reference for 'CTAB method'?
Figure 2: Increase the font size on the phylogenetic tree
Line 273: The authors declare that 'evolutionary analyses were conducted in MEGA6. At the same time, in the Section 2.5., MEGA7 was mentioned (Line 137). What MEGA version was used (or both of them?)
Line 387: 'Trichoderma' should be in Italics
Reviewer 2 Report
Dear colleagues. The article is of practical interest. However, there are questions
Line 94
"The initial screening of Trichoderma strains utilized potato dextrose agar (PDA) supplemented with triadimefon, followed by a secondary screening to optimize the concentration of cyprodinil"
Why was it necessary to use fungicides?
Line 96
"Soil samples were processed by grinding and sieving to prepare the culture medium, which was supplemented with various additives".
How was the soil culture medium prepared? Where were the soil samples taken from? What type of soil was used?
Line 147-156
«After ten days, mung bean seedling height…»,
Were the plants grown in a greenhouse? What variety, resistant or susceptible? In what conditions (temperature, lighting, humidity)?
Line 148
"Uniform-sized germinated seeds were planted in pots (dimensions: 9 cm bottom diameter × 10 cm top diameter × 12 cm height)".
How many grams of substrate were used? Which one?
Line 196-198
"Each M. robusta Rehd seedling's roots were irrigated with 300 mL of the spore suspension. Treatments included three M. robusta Rehd seedlings per group, with a control group receiving no Trichoderma spore suspension".
It is better to say that 300 ml of the code was used as a control.
Line 199
It is advisable to indicate the weather conditions.
Line 310-317
“The treatment with Trichoderma strains L7 and M19 significantly increased the activities of various defense enzymes and root vitality in M. robusta Rehd seedlings. Specifically, the activity of phenylalanine ammonia-lyase (PAL) increased by 123.16% with L7 and by 118.95% with M19. Catalase (CAT) activity saw an increase of 134.15% with L7 and 139.02% with M19. Superoxide dismutase (SOD) activity rose by 299.85% with L7 and 268.33% with M19. Additionally, root vitality increased by 161.70% with L7 and by 153.98% 315 with M19 (Fig. 4). These findings indicate that Trichoderma treatments significantly boost 316 the activity of defense enzymes (PAL, CAT, and SOD) and root vitality in M. robusta Rehd seedlings, leading to improved plant health and stress resistance”.
The data is very interesting. What methods did you use?
Line 166
“After thirty days, plant height, fresh weight, chlorophyll content, root weight, length, enzyme activity, and disease incidence were measured using modified disease grading standards”.
References are needed, especially regarding the development of the disease
Line 339
Table 2. Protective effect of Trichoderma on reseedling diseases caused by F. oxysporum HS2.
What does reseeding disease mean? How was it defined?
Reviewer 3 Report
The authors present the manuscript with the title „Screening Trichoderma strains antagonistic to Fusarium oxysporum, the pathogen of apple replant disease, and evaluating their disease prevention and growth-promoting effects”, evaluating the antagonistic properties of Trichoderma strains against Fusarium oxysporum HS2 with the aim to propose a biological control solution to decrease the need of chemical pesticides. The manuscript is well written, clear and concise, the references are appropriate and the results are interesting and useful for the field of agronomy, but also for the food safety area.
Even so, I have some comments:
1. Page 1, line 39: In my opinion, a section on Fusarium oxysporum should be introduced: toxicological data (substances produced, mechanisms of action, implications).
2. Page 1, lines 40-41: Is the term “scientific fertilization” normally used? In my opinion,, it should be defined or clearly explained.
3. Page 2, line 66: Use the abbreviation ARD, since it was already defined on line 30. Same in line 70.
4. Page 2, line 66: Some researchers have shown that…
5. Page 2, lines 70-76: Mention the references for this part.
6. Page 2, lines 70-76: End the introduction with a clear and concise aim of the study, linked to the introduction, this first section being a justification of your present study.
7. Page 2, line 79: Mention only the institution and mention the professor in the Acknowledgements section.
8. Page 2, line 82: Mention the producer for the sterile gauze.
9. Page 2, line 83: Indicate the equipment used for centrifugation, and the producer, also the conditions for centrifugation (rotations, temperature).
10. Page 2, lines 84-85: Indicate the type/model and the producer of hemocytometer.
11. Page 2, line 95: Indicate the producer for triadimefon.
12. Page 2, line 96: Indicate the equipment used for grinding.
13. Page 3, line 105: Indicate if special equipment was used (If yes, mention the type/model and the producer) or how exactly the incubation took place. Same in line 124 and 204.
14. Page 3, line 112: Indicate the equipment used for sterilization, and the producer.
15. Page 3, line 118: Indicate the model and the producer of the electronic microscope.
16. Page 3, line 127: Indicate the model and the producer of the optical microscope.
17. Page 3, line 132: Indicate the model and the producer of the PCR equipment.
18. Page 3, line 141: Mention the concentrations.
19. Page 4, lines 160-161: Mention the method used to establish chlorophyll concentration, also what were the enzymes evaluated and what method was used.
20. Page 4, line 189: Indicate the model and the producer of the fluorescence microscope.
21. Page 6, line 231: Petri
22. Page 8, line 316: Trichoderma (italic)
23. Page 12, line 405: The authors mention: Notably, the colonization M19 was higher than that of L7. Upon visual evaluation of the printed version, the graphs presented in Figure 7B, right and left, appear to be identical. Please, check the correctness of the results and the statement from lines 405-407.
24. Page 12, line 421: This study… Please be clear, the previous mentioned study or your present study? Be more specific. The same comment in lines 435, 457, 470, 472.
25. Page 12, line 482: The promotion of… Start a new paragraph from this point onwards.
26. Page 13, lines 489-493: Move these 2 phrases at the end of the Conclusion section.
27. Page 13, Conclusion section: Emphasize in 1-2 sentences the importance of the results not only at the regional level (China), but also at a global level. How can the results be transferred to other studies from other regions?
With all of these, my recommendation is to reconsider the manuscript after a Major Revision.
Round 2
Reviewer 1 Report
The revised version of the manuscript looks informative and contains some valuable data on the use of Trichoderma strains to control apple replant disease caused by F. oxysporum. I believe it can be accepted after minor modification.
Line 70: 'Trichoderma spp. controls ARD well' - this sentence should be removed.
Line 77: 'High' pathogenicity?
Line 95: 'Were'
Lines 135-136: ...By amplifying sequences...'
Reviewer 2 Report
The authors have done a great job, adding missing data. However, there are a few remarks
Line 89-94
The mycelia were filtered through four layers of gauze, which had been sterilized by high-temperature moist heat. The mycelia were filtered through four layers of sterile gauze, and after three rounds of centrifugation using an Eppendorf 5424R centrifuge at 6000 r/min and at 25℃, the spore suspension was obtained. The concentration of spores was adjusted to 1×10⁷ spores/mL using a hemocytometer (XB-K-25Plus, Shanghai Qiujing Biochemical Reagent & Instrument Co., Ltd.) and prepared for immediate use.
The phrase is duplicated.
What does three rounds mean? Was the spore suspension washed three times? How many minutes did you centrifuge?
In the future, it is better to centrifuge the spores with cooling.
Line 163
Using a syringe, 5 mL of the suspension was injected into the soil from the south and north directions.
it's better to say "on both sides"
Reviewer 3 Report
The authors present the revised manuscript with the new title „Effects of two Trichoderma strains on apple replant disease suppression and plant growth stimulation”. The revised version of the manuscript is well written, improved compared to the first one, the comments provided by the authors were enlightening, and the changes have been made accordingly.
Thus, my recommendation is to accept the manuscript in the present form.
Author Response
评论: 作者将修订后的手稿命名为“两种木霉菌株对苹果再植病害抑制和植物生长刺激的影响”。手稿的修订版写得很好,与第一版相比有所改进,作者提供的评论很有启发性,并进行了相应的更改。因此,我的建议是接受目前形式的手稿。
响应:非常感谢您的积极反馈和认可我们对手稿所做的改进。我们感谢您的深思熟虑的评论,并很高兴听到修订解决了提出的问题。感谢您的推荐,并期待稿件的最终接受。